# Higher Resistance of *Yersinia enterocolitica* in Comparison to *Yersinia pseudotuberculosis* to Antibiotics and Cinnamon, Oregano and Thyme Essential Oils

**DOI:** 10.3390/pathogens11121456

**Published:** 2022-12-01

**Authors:** Radka Hulankova

**Affiliations:** Department of Animal Origin Food and Gastronomic Sciences, Faculty of Veterinary Hygiene and Ecology, University of Veterinary Sciences Brno, 612 42 Brno, Czech Republic; hulankovar@vfu.cz; Tel.: +420-541-562-750

**Keywords:** minimum inhibitory concentration, antibiotic resistance, broth microdilution, antimicrobials, multiresistance, susceptibility

## Abstract

Yersiniosis is an important zoonotic disease; however, data are scarce on the resistance of enteropathogenic yersiniae, especially that of *Y. pseudotuberculosis*. Minimum inhibitory concentrations (MIC) of 21 antibiotics and 3 essential oils (EOs) were determined by broth microdilution for *Y. enterocolitica* bioserotype 4/O:3 strains isolated from domestic swine (*n* = 132) and *Y. pseudotuberculosis* strains isolated from wild boars (*n* = 46). For 15 of 21 antibiotics, statistically significant differences were found between MIC values of *Y. enterocolitica* and *Y. pseudotuberculosis*. While *Y. enterocolitica* was more resistant to amoxiclav, ampicillin, cefotaxime, cefuroxime, gentamicin, imipenem, meropenem, tetracycline, tobramycin, and trimethoprim, *Y. pseudotuberculosis* was more resistant to cefepime, ceftazidime, colistin, erythromycin, and nitrofurantoin. Statistically significant differences were found between various essential oils (*p* < 0.001) and species (*p* < 0.001). The lowest MICs for multiresistant *Y. enterocolitica* (*n* = 12) and *Y. pseudotuberculosis* (*n* = 12) were obtained for cinnamon (median 414 and 207 μg/mL, respectively) and oregano EOs (median 379 and 284 μg/mL), whereas thyme EO showed significantly higher MIC values (median 738 and 553 μg/mL; *p* < 0.001). There was no difference between *Y. enterocolitica* strains of plant (1A) and animal (4/O:3) origin (*p* = 0.855). The results show that *Y. enterocolitica* is generally more resistant to antimicrobials than *Y. pseudotuberculosis*.

## 1. Introduction

Yersiniosis is an important zoonotic disease that spreads mainly via alimentary transmission. In 2020, 5668 confirmed cases of yersiniosis were reported in 25 EU countries. The overall rate was 1.8 cases per 100,000 population with 29% hospitalisation and 0.07% mortality. In the past decades, the highest rates have been reported in Scandinavian countries, Baltic states, Czechia, and Slovakia. In 2020, yersiniosis was the third most commonly reported foodborne zoonotic disease in the EU/EEA, which proves its importance. The disease can be caused by *Yersinia enterocolitica* or *Y. pseudotuberculosis*, although *Y. enterocolitica* causes the majority (99%) of human infections in the EU [1]. However, outbreaks caused by *Y. pseudotuberculosis* have been reported in France, Japan, Russia, and Scandinavia [2]. Domestic swine are considered the main reservoir of yersiniae for humans, especially *Y. enterocolitica* bioserotype 4/O:3, as they are asymptotic carriers of yersiniae in the tonsils, gut, and associated lymph nodes. However, wild animals, particularly wild boars, can also be vectors, especially for hunters [3]. What’s more, *Yersinia* is well adapted to low temperatures and can replicate even during refrigerated storage, with its numbers eventually reaching the infectious dose. This presents a serious risk to consumers [3,4].

The increasing occurrence of antibiotic resistance is a global problem. Although most infections are self-limiting, antibiotics are used to treat more severe forms of human yersiniosis, such as enterocolitis, in immunodeficient individuals or patients with septicaemia. In general, ciprofloxacin is recommended for enterocolitis and a combination of two antibiotics is recommended for more severe forms, e.g., a combination of an aminoglycoside, such as gentamicin, with a 3rd generation cephalosporin or ciprofloxacin. Other antibacterial drugs, e.g., tetracyclines, chloramphenicol, cotrimoxazole, or trimethoprim-sulfamethoxazole can be also potentially used for treatment [4,5]. Doxycycline is also used to treat prolonged deep-tissue infections [6]. Regarding antibiotic resistance, *Yersinia* species, especially *Y. enterocolitica*, produce chromosomally encoded beta-lactamases that confer resistance to all beta-lactam antibiotics, such as ampicillin, penicillin, and 1st generation cephalosporins [3,4]. Furthermore, natural resistance to macrolides (erythromycin) based on the efflux pump principle has been reported [4]. These findings have been confirmed in *Y. enterocolitica* by many studies, and have been summarised in several review articles [4,7]. It is believed that resistant isolates found in humans likely originate from the animal environment in which antimicrobials or similar drugs had been used or are in use in veterinary care, e.g., apramycin, chloramphenicol, florphenicol, quinolones, streptomycin, tetracycline, and thiamphenicol [4].

Essential oils (EOs) are volatile, aromatic phytochemicals with analgesic, antioxidant, anti-inflammatory, antimicrobial, and antiseptic properties. They lack the negative side effects of synthetic drugs and there has been no proof of acquired resistance of microorganisms to EOs to date. In addition, synergistic effects of various EOs with several conventional synthetic drugs have been reported [8].

The antimicrobial properties of essential oils have been studied extensively in vitro for several decades. However, the majority of studies have focused on well-known food pathogens such as *Salmonella* spp., *Listeria monocytogenes*, or shiga toxin-producing *Escherichia coli*. Studies investigating enteropathogenic yersiniae, especially *Y. pseudotuberculosis*, are scarce and the same is true for studies reporting their antimicrobial resistance.

The aim of this study was to assess the efficiency of common antibiotics and essential oils against strains of *Y. enterocolitica* and *Y. pseudotuberculosis* originating from animal and plant sources.

## 2. Materials and Methods

### 2.1. Yersinia Strains

*Yersinia enterocolitica* (bioserotype 4/O:3, *n* = 132, isolated from tonsils of slaughtered pigs) and *Yersinia pseudotuberculosis* (serotype not determined, *n* = 46, isolated from tonsils of wild boar) were used in this study for determination of antibiotic resistance. The strains were isolated from tonsils according to modified ISO 10273, as described in an Italian study [9]. In short, the method included direct plating on cefsulodin–irgasan–novobiocin (CIN) agar and cold enrichment in PSB broth (M941, Himedia, Mumbai, India) with 1% mannitol (Himedia, Mumbai, India) and 0.15% bile salts (Himedia, Mumbai, India), followed by alkali treatment, and plating on CIN agar (M843, Himedia, Mumbai, India). Species identification and typing was performed using the VITEK2^®^ (bioMérieux, Marcy-l’Étoile, France) and ENTEROTEST 24N (Erba-Lachema, Brno, Czechia) identification system and by sera for slide agglutination (SIFIN, Berlin, Germany).

In the second part of this study, 12 strains of each species, which were resistant to multiple antibiotics, were chosen for determination of their sensitivity to EOs. Simultaneously, 12 multiresistant strains of *Y. enterocolitica* isolated from vegetables were tested. The strains of plant origin were isolated according to ISO 10273:2003 from fresh and frozen vegetables during a previous study at Veterinary Research Institute, Czech Republic [10] and belonged to biotype 1A (serotype O:8 and O:5). The number of antimicrobials to which each strain was resistant is described in Table 1.

### 2.2. Essential Oils

Commercial EOs from cinnamon (*Cinnamomum zeylanicum*, Indonesia), oregano (*Origanum vulgare*, Spain), and thyme (*Thymus vulgare*, Spain) were purchased from Nobilis Tilia, Krásná Lípa, Czechia. The specific chemical composition of each oil batch was determined by GC-MS in an accredited laboratory in Germany (where the oils were manufactured, by Joh. Vögele KG, Lauffen am Neckar) and is available in Appendix A (Appendix A).

### 2.3. Determination of Minimum Inhibitory Concentration of Antibiotics

Strains used in this study were tested using the VITEK2^®^ system (bioMérieux, Marcy-l’Étoile, France), based on broth microdilution, using AST-N199 test cards (bioMérieux, Marcy-l’Étoile, France). The panel of 17 antimicrobials included ampicillin, amoxicillin/clavulanic acid, cefuroxime, cefoxitin, cefotaxime, ceftazidime, cefepime, imipenem, meropenem, gentamicin, tobramycin, ciprofloxacin, norfloxacin, nitrofurantoin, colistin, trimethoprim, and trimethoprim/sulfamethoxazole. In addition, resistance to erythrofloxacin, streptomycin, tetracycline, and chloramphenicol (Sigma-Aldrich, Taufkirchen, Germany) was determined by the dilution method in Mueller Hinton Broth (CM0405, Oxoid, Basingstoke, UK) according to EUCAST standards [11]. The susceptibility/resistance was interpreted according to MIC breakpoints for Enterobacterales published by the European Committee on Antimicrobial Susceptibility Testing [6].

### 2.4. Determination of Minimum Inhibitory Concentration of Essential Oils

Minimum inhibitory concentration (MIC) was determined by the broth microdilution method with two replications, as previously described [12], using tryptone soya broth (TSB, CM0129, Oxoid, Basingstoke, UK) and cultivation at 30 °C/24 h. The concentration range was 0–0.12% (*v/v*) and the results were expressed in μg/mL taking into account the density of each oil batch (see Appendix A).

### 2.5. Statistical Analysis

Statistical analysis was performed using Statistica v. 7.1 software (StatSoft, Tulsa, OK, USA). MIC values for *Y. enterocolitica* and *Y. pseudotuberculosis* were individually compared for each antibiotic using the non-parametric Mann-Whitney U test, and frequencies of resistant/intermediate/susceptible strains were evaluated using the Chi-Square statistic. Data for EOs were analysed using the open-source statistical software R version 4.0.0 [13]. An ordered regression model was applied using the “ordinal” package in R [14], with measurement number and strain as the random effects and group (*Y. enterocolitica* 1A, *Y. enterocolitica* 4/O:3, *Y. pseudotuberculosis*) and EO type as the fixed effects. Comparisons were performed using the “emmeans” package in R with Holm-Bonferroni’s correction [15]. A P level of 0.05 was set as statistically significant.

## 3. Results and Discussion

### 3.1. Antibiotic Resistance

For 15 of 21 antibiotics, statistically significant differences were found between MIC values of *Y. enterocolitica* and *Y. pseudotuberculosis* strains (Table 2). While *Y. enterocolitica* was more resistant to amoxiclav, ampicillin, cefotaxime, cefuroxime, gentamicin, imipenem, meropenem, tetracycline, tobramycin, and trimethoprim, *Y. pseudotuberculosis* was more resistant to cefepime, ceftazidime, colistin, erythromycin, and nitrofurantoin. These differences were also partially reflected by significant differences in the proportion of resistant and sensitive strains (Figure 1), especially for ampicillin, colistin, and tetracycline (*p* < 0.001). Sensitivity/resistance to streptomycin and erythromycin was not evaluated as there are currently no EUCAST or CLSI breakpoint values for MIC. All tested strains were sensitive to ciprofloxacin, imipenem, tobramycin, trimethoprim, and sulfamethoxazole potentiated by trimethoprim, while only one strain was resistant to norfloxacin (*Y. enterocolitica*, MIC 16 mg/L). Cefepime and meropenem showed only intermediate resistance and a very low proportion of strains (2%) were resistant to nitrofurantoin.

Only 2.3% of *Y. enterocolitica* isolates, but 12.5% of *Y. pseudotuberculosis* isolates, were resistant to chloramphenicol (*p* = 0.007). The current occurrence of resistance to chloramphenicol, which has been banned in food animals since 1994, is usually explained by the frequent use of structurally similar drugs, such as thiamphenicol and florfenicol, on farms [9]. The antimicrobial resistance of isolates from wild boar could be associated with its transfer between strains present in both domestic swine and wild boar, and with the overpopulation of wild boar since the animals are thus more frequently in contact with pigs and waste materials [16]. Most studies have reported resistance of *Y. enterocolitica* to chloramphenicol from 0 to 4% [4]; however, higher values of 53% and 38% have been reported for porcine isolates [9,17]. However, our study shows that the resistance is more spread among *Y. pseudotuberculosis* in Czech wild boar population, although *Y. pseudotuberculosis* isolates from wild boars in neighboring Germany showed no resistance [18]. The gene variants encoding chloramphenicol acetyl transferase have been found previously in *Y. pseudotuberculosis* isolates [2].

All *Y. pseudotuberculosis* isolates were sensitive to tetracycline (MIC_90_ 2 mg/L), but one third of *Y. enterocolitica* strains were resistant (MIC_90_ 8 mg/L; *p* < 0.001). Tetracycline resistance can be used to estimate the effectiveness of doxycycline, which is used in the treatment of *Yersinia* infections; the MIC limit is ≤ 4 mg/L for wild-type strains [11]. High *Y. enterocolitica* resistance to tetracyclines (20% and 50% of strains) was also noted in isolates from the tonsils of fattening pigs in Italy [9,17]. Furthermore, a higher percentage of resistance has been reported for porcine isolates of *Y. enterocolitica* comparison to that of *Y. pseudotuberculosis*, namely 8.4% versus 0% [19] and 1% versus 0% [20,21]. In comparison to a previous study that our laboratory performed between 2005 and 2007, resistance of *Y. enterocolitica* to tetracycline in fattening pigs has increased during the past 20 years from 13% to 35% [22].

Differences between the two species were also evident in their susceptibility to ampicillin. *Y. enterocolitica* was more frequently resistant to ampicillin (98% of isolates) than *Y. pseudotuberculosis* (13%, *p* < 0.001). A similar disproportion was found in other studies. The proportion of ampicillin-resistant strains of *Y. enterocolitica* and *Y. pseudotuberculosis* in a Latvian study [20] was 100% and 0%, respectively, and 68.7% and 3.6% in a Greek study [19], respectively. No ampicillin-resistant *Y. pseudotuberculosis* strains were found in wild boars in Germany [18]. On the other hand, for isolates of plant origin tested in Czechia, no difference between the two species was found, with 100% resistance to ampicillin in all pathogenic and non-pathogenic species included in the study [10.] An even larger disproportion was found in the resistance to colistin (polymyxin E), with 3% of *Y. enterocolitica* strains being resistant and the same percentage being susceptible among *Y. pseudotuberculosis* strains (Figure 1). A previous in vitro study reported increased resistance of *Y. pseudotuberculosis* to polymyxin at 37 °C compared to *Y. enterocolitica* [23]. High colistin resistance (90%) among *Y. pseudotuberculosis* isolates from wild boars was also reported in Germany [18].

The differences between *Y. enterocolitica* and *Y. pseudotuberculosis* were also evident when comparing the most common phenotypes of resistance (Table 3). The most common phenotype of *Y. enterocolitica* was single resistance to ampicillin (23%) and its combinations with other drugs, whereas the most common phenotype for *Y. pseudotuberculosis* was single resistance to colistin (11%) and its combinations. Multidrug resistant strains of *Y. enterocolitica* and *Y. pseudotuberculosis* (up to 6 antibiotic families) have been detected in previous studies [2,7]. Genes encoding multiple resistance can be acquired via large plasmids that are widespread among Enterobacteriaceae bacteria [2].

### 3.2. Essential Oils

Statistically significant differences were found between various EOs (*p* < 0.001) and species (*p* < 0.001); however, no interaction was detected (*p* = 0.106). The lowest MICs (Table 4) were obtained for cinnamon and oregano EOs, whereas thyme EO showed significantly higher MIC values (*p* < 0.001). The MIC values of *Y. enterocolitica* were very slightly lower (median 310–414 μg/mL) than those of other pathogens (*Salmonella enteritidis*, *Escherichia coli* O157, *Listeria monocytogenes*, and *Staphylococcus aureus*; median 414–569 μg/mL) when using the same assay and oregano and cinnamon EOs [12]. Antimicrobial activity of different EOs against *Y. enterocolitica* has been recently reviewed and oregano, rosemary, thyme, and basil were identified as the most promising EOs deserving further studies [24]. Only one study focused on both oregano and thyme EOs for in vitro inhibition of *Y. enterocolitica*, with a higher MIC value for thyme EO (2.34 mg/mL) than for oregano EO (0.59 mg/mL) [25]. The difference in MIC values between these EOs and those in our study may result from a lower content of oxygenated monoterpenes (e.g., carvacrol and thymol) in thyme EO (55% in our study) compared to that in oregano EO (75% in our study).

*Y. pseudotuberculosis* was significantly (*p* < 0.001) less resistant to EOs than *Y. enterocolitica*. This finding is clearly species related, as there was no difference between *Y. enterocolitica* strains of plant and animal origin (*p* = 0.855), which were 1A and 4 biotype strains, respectively. Biotype 1A strains, for a long time generally regarded as non-pathogenic, are now considered emerging human pathogens [3]. As mentioned before, data on EO efficacy against *Y. pseudotuberculosis* are scarce. In 2022, less than 20 articles pertaining to the effect of EOs on *Y. pseudotuberculosis* were available in the Web of Science database, and they were focused on less common sources of EOs (*Anthemis* spp., *Campanula olympica*, *Caucasalia macrophylla, Inula thapsoides, Omphalodes cappadocica, Myosotis alpestris, Myrtus nivellei, Paeonia mascula, Rhododendrum caucasium, Rumex crispus, Salvia staminea, Satureja hortensis,* and *Thamnobryum alopecurum*). Only one publication pertained to both *Y. enterocolitica* and *Y. pseudotuberculosis*: the MIC of EO from *Satureja hortensis* was the same (7.81 μg/mL) for both species, but only one clinical isolate per species was used in the study [26].

The resistance against EOs was not correlated with resistance against antibiotics. Several mechanisms of action have been described for EOs, among which the main mechanisms include increased permeabilization of the cytoplasmic membrane of bacteria and inhibition of ATP synthesis [27], whereas antibiotics predominantly inhibit the synthesis of proteins and nucleic acids. The mechanism of resistance to antibiotics can be mediated by enzymatic degradation, alteration of drug targets, decreased uptake, or increased efflux [8]. The MexAB-OprM efflux pump can reportedly naturally protect *Pseudomonas aeruginosa* against both antibiotics and phenolic compounds of EOs, namely carvacrol [28]. However, results of several studies suggest that resistance to antibiotics does not confer cross-resistance to EOs; in fact, there is some evidence that EOs and their components can interfere with antibiotic resistance mechanisms and act synergistically with antibiotics [8,27]. Development of combination therapies using the synergy between EOs or their components and antibiotics could be a promising approach to combat the increasing resistance to antibiotics [27]. On the other hand, the potential use of EOs in therapy and food protection has several potential drawbacks that should be mentioned, including variability in chemical composition within batches of EOs of the same botanical origin and the price of EOs. Although the results of in vitro studies may prove promising, the potential use of EOs to inhibit *Yersinia* or other pathogens in minced pork or on pork cuts may be hindered, for example, by the high fat content of meat [29]. The required inhibitory concentration may be high enough to result in unacceptably strong herbal/spicy aromas [30]. Further studies investigating the mechanisms of action of EOs in food, especially at low temperatures and under specific conditions such as vacuum or modified atmosphere, should be encouraged.

## 4. Conclusions

Enteropathogenic yersiniae showed a high level of resistance to some penicillins (ampicillin), but good susceptibility to carbapenems, quinolones, and sulfonamides. Cefoxitin (2nd generation) was the least effective cephalosporin against *Y. enterocolitica* and *Y. pseudotuberculosis* in this study, whereas cefepime (4th generation) was the most effective. As for aminoglycosides, tobramycin seems to be more fitting for treatment than gentamicin. *Y. enterocolitica* was generally more resistant than *Y. pseudotuberculosis*, which was, on the other hand, much more resistant to colistin. EOs from cinnamon, oregano, and thyme (to a lesser extent) were effective against both *Yersinia* species, with *Y. enterocolitica* being more resistant than *Y. pseudotuberculosis*. Therefore, EOs can be a promising alternative for the inhibition of *Yersinia* strains that display multiresistance to conventional antibiotics.

## Figures and Tables

**Figure 1 pathogens-11-01456-f001:**
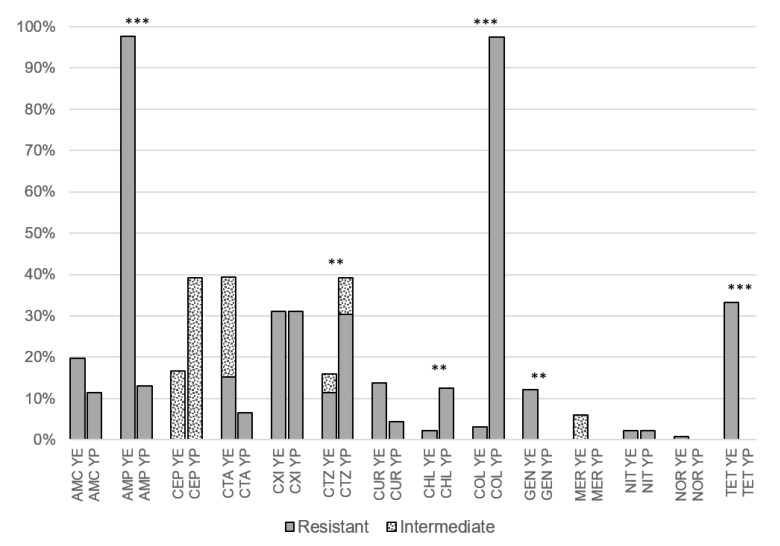
Proportion of resistance and intermediate sensitivity among porcine isolates of *Yersinia enterocolitica* (YE, *n* = 132) and *Yersinia pseudotuberculosis* (YP, *n* = 46). AMC, amoxiclav; AMP, ampicillin; CEP, cefepime; CTA, cefotaxime; CXI, cefoxitin; CTZ, ceftazidime; CUR, cefuroxime; CHL, chloramphenicol; COL, colistin; GEN, gentamicin; MER, meropenem; NIT, nitrofurantoin; NOR, norfloxacin; PIT, piperacillin/tazobactam; TET, tetracycline. The asterisk indicates a statistically significant difference between *Y. enterocolitica* and *Y. pseudotuberculosis* in the proportion of resistant isolates (** *p* < 0.01, *** *p* < 0.001).

**Table 1 pathogens-11-01456-t001:** Multiresistant strains used for determination of efficacy of essential oils (*n* = 36).

Species	Bioserotype	Origin	Number of Resistances	Number of Strains
*Y. enterocolitica*	1A/O:5, O:8	Vegetables	6	1
			5	1
			4	8
			3	2
*Y. enterocolitica*	4/O:3	Domestic swine	8	1
			6	5
			5	6
*Y. pseudotuberculosis*	-	Wild boar	5	2
			4	5
			3	5

**Table 2 pathogens-11-01456-t002:** Minimum inhibitory concentrations (MIC, in mg/L) of antibiotics for *Yersinia enterocolitica* (*n* = 132) and *Yersinia pseudotuberculosis* (*n* = 46).

Antibiotic	*Y. enterocolitica*			*Y. pseudotuberculosis*			*p* Level
	**MIC_90_ ***	**MIC_50_ ^†^**	**Min–Max ^§^**	**MIC_90_**	**MIC_50_**	**Min–Max**	
Amoxiclav	16	2	2–32	16	2	2–32	**0.011**
Ampicillin	32	32	4–32	12	2	2–16	**<0.001**
Cefepime	4	1	1–4	2	1	1–4	**0.017**
Cefotaxime	4	1	1–8	1	1	1–4	**<0.001**
Cefoxitin	32	8	4–66	32	4	4–64	0.134
Ceftazidime	8	1	1–64	32	1	1–64	**0.002**
Cefuroxime	16	8	2–64	8	2	1–32	**<0.001**
Chloramphenicol	8	4	1–16	16	4	4–16	0.575
Ciprofloxacin	0.25	0.25	0.025–0.5	0.25	0.25	0.25–0.25	0.552
Colistin	0.5	0.5	0.5–16	16	16	0.5–16	**<0.001**
Erythromycin	64	32	4–128	64	64	32–64	**<0.001**
Gentamicin	4	1	1–4	1	1	1–1	**<0.001**
Imipenem	0.475	0.25	0.25–0.5	0.25	0.25	0.25–0.25	**0.039**
Meropenem	1	0.25	0.25–4	0.25	0.25	0.25–0.25	**<0.001**
Nitrofurantoin	64	32	16–128	64	64	16–128	**0.001**
Norfloxacin	0.5	0.5	0.5–16	0.5	0.5	0.5–0.5	0.564
Streptomycin	16	8	1–32	64	8	2–64	0.242
Tetracycline	8	4	0.5–16	2.2	2	1–4	**<0.001**
Tobramycin	2	1	1–2	1	1	1–1	**0.008**
Trimethoprim	2	2	0.5–4	1	1	0.5–2	**<0.001**
Trimethoprim–Sulfamethoxazole	20	20	20–20	20	20	20–20	1.000

***** MIC_90_ the lowest concentration of the antibiotic at which 90 % of the isolates were inhibited; **^†^** MIC_50_ the lowest concentration of the antibiotic at which 50 % of the isolates were inhibited; **^§^** min-max all the values are from ≤ up to ≥.

**Table 3 pathogens-11-01456-t003:** Resistance patterns of *Yersinia enterocolitica* (*n* = 132) and *Yersinia pseudotuberculosis* (*n* = 46) isolates.

Phenotype	*Y. enterocolitica*		Phenotype	*Y. pseudotuberculosis*	
	R *	N ^†^		R	N
Amp	1	30 (22.7%)	Col	1	14 (10.6%)
AmpTet	2	22 (16.7%)	ChlCol	2	6 (4.5%)
AmpCxi	2	14 (10.6%)	CxiCol	2	4 (3.0%)
AmcAmpCxi	3	9 (6.8%)	AmcCol	2	3 (2.3%)
AmpGen	2	7 (5.3%)	AmcAmpCxiCol	4	2 (1.5%)
AmpGenTet	3	6 (4.5%)	AmpCxiCtzCol	4	2 (1.5%)
AmcAmp	2	4 (3.0%)	CtaCtz	2	2 (1.5%)
AmpCtaCtzCurGenTet	6	3 (2.3%)	AmpCxiCtzCurCol	5	1 (0.8%)
AmpCtaCtzCurTet	5	3 (2.3%)	CtaCxiCtzCurCol	5	1 (0.8%)
AmcAmpCtaCxi	4	3 (2.3%)	CxiCtzChlCol	4	1 (0.8%)
AmpCxiTet	3	3 (2.3%)	AmpCtzCol	3	1 (0.8%)
AmpCur	2	3 (2.3%)	CxiCtzCol	3	1 (0.8%)
AmcAmpCxiCur	4	2 (1.5%)	CxiCtzNit	3	1 (0.8%)
AmcAmpCxiNit	4	2 (1.5%)	CxiChlCol	3	1 (0.8%)
AmpCurTet	3	2 (1.5%)	CtzChlCol	3	1 (0.8%)
AmpCta	2	2 (1.5%)	CxiCtz	2	1 (0.8%)
AmcAmpCtaCxiCtzCurChlNit	8	1 (0.8%)	CtzCol	2	1 (0.8%)
AmcAmpCtaCxiCtzCol	6	1 (0.8%)	Ctz	1	1 (0.8%)
AmpCtaCxiCtzCurTet	6	1 (0.8%)	Chl	1	1 (0.8%)
AmcAmpCtaCxiChl	5	1 (0.8%)	sensitive	0	1 (0.8%)
AmpCtaCxiCtzCur	5	1 (0.8%)			
AmpCtaCxiCtzCol	5	1 (0.8%)			
AmpCtaCtzTet	4	1 (0.8%)			
AmcAmpCur	3	1 (0.8%)			
AmcAmpChl	3	1 (0.8%)			
AmpCtaCtz	3	1 (0.8%)			
AmpCtaTet	3	1 (0.8%)			
AmpCxiCol	3	1 (0.8%)			
AmcCxiTet	3	1 (0.8%)			
AmpCtzCur	3	1 (0.8%)			
AmpNorTet	3	1 (0.8%)			
CtzCol	2	1 (0.8%)			
sensitive	0	1 (0.8%)			

* R, number of resistances; ^†^ N, number of strains; AMC, amoxiclav; AMP, ampicillin; CTA, cefotaxime; CXI, cefoxitin; CTZ, ceftazidime; CUR, cefuroxime; CHL, chloramphenicol; COL, colistin; GEN, gentamicin; NIT, nitrofurantoin; NOR, norfloxacin; TET, tetracycline.

**Table 4 pathogens-11-01456-t004:** Minimum inhibitory concentrations (MIC, in μg/mL) of essential oils for *Yersinia enterocolitica* (YE) and *Yersinia pseudotuberculosis* (YP).

	YE 1A (*n* = 12)	YE 4/O:3 (*n* = 12)	YP (*n* = 12)
	median [range]	median [range]	median [range]
Origin	vegetable	domestic swine	wild boar
Cinnamon	414 ^Aa^ [207; 517]	310 ^Aa^ [207; 414]	207 ^Ab^ [103; 414]
Oregano	379 ^Aa^ [284; 474]	379 ^Aa^ [284; 474]	284 ^Ab^ [190; 474]
Thyme	738 ^Ba^ [553; 922]	738 ^Ba^ [553; 922]	553 ^Bb^ [369; 738]

^a-b^ mark statistically significant differences within a row; ^A-B^ mark statistically significant differences within a column.

## Data Availability

The data presented in this study are openly available in Mendeley at https://data.mendeley.com/datasets/8tskwsbt3h/1, doi:10.17632/8tskwsbt3h.1.

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
