# Peer review of "Higher Resistance of Yersinia enterocolitica in Comparison to Yersinia pseudotuberculosis to Antibiotics and Cinnamon, Oregano and Thyme Essential Oils"

_pathogens, 2022, doi:10.3390/pathogens11121456_

Round 1

Reviewer 1 Report

Brief summary :

This paper aims to assess the resistance of Yersinia enterocolitica and Yersinia pseudotuberculosis strains to antibiotics and essential oils.

The author tested 21 antibiotics on 178 strains. Antibiotic resistance patterns in Y. enterocolitica were different from those of Y. pseudotuberculosis, and multi-resistance patterns were found in the 2 species.

The microbial activity of cinnamon, oregano and thyme essential oils were tested on 36 multi-drug resistant strains of Y. enterocolitica and Y. pseudotuberculosis. The lowest MIC were obtained with cinnamon and oregano.

General comments:

-       The manuscript is well written, very interesting and promising for the development of novel antimicrobial therapies.

-       There is no co-author. Is the author actually the only contributor of the study?

-       It is unclear whether the author tested a control strain for the determination of MIC of essential oils. For instance, a reference strain with known values of MIC for the protocol used in this study. I can’t see any results for a control sample.

-        The perspective should be more detailed: steps to go through before using essential oils

o   in combination with antibiotics to treat infections.  

o   in food chain to prevent the development of the bacteria (Yersinia)

Introduction:

The ability of Yersinia cells to grow at 4°C should be mentioned as it increases the risk of human infection. Yersinia may develop in contaminated food stored a few days at cold temperatures.

Results:

-       The section title should be “Results and Discussion”.

-       Lines 215-222: it is unclear whether there are 15 articles about efficacy of essential oils on Y. pseudotuberculosis and 1 article on efficacy of essential oil on Y. pseudotuberculosis and Y. enterocolitica. Please, clarify this point and remove the sentence “- as far as I know, at the time of the submission of this manuscript, only 15 articles were available at the Web of Science database”.

Reviewer 2 Report

Please consider these comments:

1- The title should be modified.

2- The abstract should be rewritten (introduction, methods, results, and discussion)

3- Other keywords that are representative of the manuscript should be added

4-The values of the MICs of the essential oils should be added.

5-Molecular mechanism of action of the essential oils should be studied

Reviewer 3 Report

In the manuscript, the author investigated the resistance of Yersinia enterocolitica (Ye) and Y. pseudotuberculosis (Yptb) isolates to antibiotics and some essential oils (EOs). The author recognized that Yersiniosis is an important zoonotic disease, and domestic swine are considered the main reservoir of yersiniae for humans. Though wild boars can also be a vector. The author used 132 Ye boserotype 4/O:3 isolates from slaughtered pigs and 46 Yptb isolates from wild boars to determine antibiotic resistance. Next, the author used 12 strains from each species that are resistant to multiple antibiotics to determine their sensitivity to EOs. Another 12 multi-resistant strains of Ye (biotype 1A, serotype O:8 and O:5) isolated from vegetables were tested for EO resistance. The manuscript is well organized; however, it is not always clear what were compared when the statistical analysis described. In addition, the findings can be strengthened with additional association studies and discussions. Suggestions to improve the manuscript is listed below. 

Main points:

1.     Please clarify what were compared. On page 3 lines 125-126, it reads that “For 15 of 21 antibiotics, statistically significant differences were found between MIC of Y. enterocolitica and Y. pseudotuberculosis strains (Table 2).” Does this mean the comparison is between the MICs of Ye and Yptb for a specific antibiotic? But in Table 2, for example, of Amoxiclav, all the values listed are the same between Ye and Yptb. 

2.     Similarly, what were compared are not clearly described on page 6 lines 193-194 as well.

3.     Related, if the percentage of Ye resistant to a certain antibiotic is significantly different from the percentage of Yptb to the same antibiotic, would the author please elaborate on the implication of such a finding? Afterall, the source of Ye and Yptb are quite different. To compare to the percentage of antibiotic resistance, is it possible to reference to other bacterial pathogens from the same source? For example, the majority of Ye are ampicillin-resistant, is this specific for Ye or the same is true for other pathogens that can be isolated from domestic pigs?

4.     Is there any association between the presence of virulence determinants and resistance to antibiotic/EO? For example, any association between the presence of the virulence plasmid and antibiotic resistance?

Minor points:

1.     Please spell out CIN agar.

2.     Please provide reference to “Yersinia produce chromosomally encoded beta-lactamases that make it resistant to all beta-lactam antibiotics such as ampicillin, penicillin and 1st generation cephalosporins” on page 2, lines 47-48. Please also indicate if this is for Ye or for all pathogenic Yersinia species.

Round 2

Reviewer 1 Report

I think that you did not understand well my remark. Reference strains are used in MIC determination of antibiotics in order to make sure that the experimental conditions are right (whatever the species of the reference strain). It should be the same for the determination of MIC of EOs.

Reviewer 2 Report

The manuscript can be accepted.